# Nrf2 Deficiency Exacerbates the Decline in Swallowing and Respiratory Muscle Mass and Function in Mice with Aspiration Pneumonia

**DOI:** 10.3390/ijms252111829

**Published:** 2024-11-04

**Authors:** Hikaru Hashimoto, Tatsuma Okazaki, Yohei Honkura, Yuzhuo Ren, Peerada Ngamsnae, Takuma Hisaoka, Yasutoshi Koshiba, Jun Suzuki, Satoru Ebihara, Yukio Katori

**Affiliations:** 1Department of Otolaryngology-Head and Neck Surgery, Tohoku University Graduate School of Medicine, 1-1 Seiryo-machi, Aoba-ku, Sendai 980-8574, Japan; htdnk259@yahoo.co.jp (H.H.);; 2Department of Rehabilitation Medicine, Tohoku University Graduate School of Medicine, 1-1 Seiryo-machi, Aoba-ku, Sendai 980-8574, Japansatoru.ebihara.c4@tohoku.ac.jp (S.E.); 3Center for Dysphagia of Tohoku University Hospital, 1-1 Seiryo-machi, Aoba-ku, Sendai 980-8574, Japan

**Keywords:** aspiration pneumonia, Nrf2, swallowing muscles, respiratory muscles, muscle atrophy

## Abstract

Aspiration pneumonia exacerbates swallowing and respiratory muscle atrophy. It induces respiratory muscle atrophy through three steps: proinflammatory cytokine production, caspase-3 and calpain, and then ubiquitin–proteasome activations. In addition, autophagy induces swallowing muscle atrophy. *Nrf2* is the central detoxifying and antioxidant gene whose function in aspiration pneumonia is unclear. We explored the role of Nrf2 in aspiration pneumonia by examining swallowing and respiratory muscle mass and function using wild-type and *Nrf2*-knockout mice. Pepsin and lipopolysaccharide aspiration challenges caused aspiration pneumonia. The swallowing (digastric muscles) and respiratory (diaphragm) muscles were isolated. Quantitative RT-PCR and Western blotting were used to assess their proteolysis cascade. Pathological and videofluoroscopic examinations evaluated atrophy and swallowing function, respectively. *Nrf2*-knockouts showed exacerbated aspiration pneumonia compared with wild-types. *Nrf2*-knockouts exhibited more persistent and intense proinflammatory cytokine elevation than wild-types. In both mice, the challenge activated calpains and caspase-3 in the diaphragm but not in the digastric muscles. The digastric muscles showed extended autophagy activation in *Nrf2*-knockouts compared to wild-types. The diaphragms exhibited autophagy activation only in *Nrf2*-knockouts. *Nrf2*-knockouts showed worsened muscle atrophies and swallowing function compared with wild-types. Thus, activation of Nrf2 may alleviate inflammation, muscle atrophy, and function in aspiration pneumonia, a major health problem for the aging population, and may become a therapeutic target.

## 1. Introduction

Among the chief causes of mortality, the combination of pneumonia and aspiration pneumonia ranks as third or fourth in Japan [1]. Especially in older people, impaired reflexes of swallowing and coughing are two critical factors that increase the risk for developing these diseases [2].

The reductions in muscle strength, size, and performance associated with aging are called sarcopenia [3]. Recent advances in sarcopenia research suggest its association with pneumonia in older people [4]. In 2019, a position paper suggested that the swallowing-related muscles suffering from sarcopenia could induce swallowing dysfunction [5]. The intensity of respiratory muscle force regulates the coughing force. Indeed, in older people, weakened respiratory muscle strength increased the risk of pneumonia development and death [6,7].

Aspiration pneumonia causes acute and chronic inflammation [8]. The inflammation induces muscle atrophy by exposure to proinflammatory cytokines [9]. Proinflammatory cytokines initially cleave myofibrillar proteins via calpains and caspase-3 [10]. Then, the ubiquitin–proteasome pathway breaks down the cleaved myofibrils. Activation of calpain was determined by observing the breakdown of the fodrin protein, resulting in 145–150 kDa bands. Activation of caspase-3 was determined by the fodrin cleavage into a 120 kDa band [9]. The E3 ubiquitin ligases constitute a part of the ubiquitin–proteasome cascade. The E3 ubiquitin ligases atrogin-1 and Muscle RING-finger protein 1 (MuRF-1) are specific to muscles [11]. As another pathway, autophagy causes muscle atrophy in hypoxia and inflammation [12,13]. Oxidative stress was linked to the muscle proteolysis pathway via autophagy [12].

Nuclear factor erythroid 2-related factor 2 (Nrf2) controls various gene expressions and regulates responses to oxidative stress and protection [14,15,16,17]. When encountering oxidative and electrophilic stressors, Nrf2 is activated systemically as a central player of many genes involved in detoxification and antioxidation [18,19,20]. The Kelch-like ECH-associated protein 1 (KEAP1)–Cullin3 ubiquitin E3 ligase complex ubiquitinates Nrf2, which undergoes degradation under normal conditions. Exposure to reactive oxygen species or electrophiles inactivates KEAP1. Subsequently, Nrf2 undergoes stabilization and migrates to the nucleus, initiating the activation of numerous genes associated with cellular differentiation, protection, and proliferation [21,22,23,24]. Accordingly, Nrf2 plays important roles in cancers [25,26,27], especially as a target of susceptibility to chemotherapy and radiotherapy [28,29,30]. In addition, Nrf2 plays potent anti-inflammatory roles, including in the lungs [31,32,33]. Properly activated Nrf2 prevents and alleviates numerous pathological conditions, including chronic inflammation [34,35]. In skeletal muscles, Nrf2 is involved in autophagy, regeneration, exercise capacity, and muscle mass loss [34,36,37,38,39,40]. In *Nrf2*-knockout mice, the administration of the bacteria-derived toxin LPS intratracheally exacerbated inflammation in the lungs [41]. In addition, Nrf2 contributes to the anti-aging process [42]. Nrf2 especially inhibited the aging-related decline in salivary gland health, which plays a significant role in preventing aspiration pneumonia [43]. In the skeletal muscles, decreases in contractile force and muscle mass were accelerated in aged *Nrf2*-knockout mice [40].

The videofluoroscopic examination of swallowing evaluates swallowing impairment in humans [44]. Recent studies have developed it and applied it to mice [45]. A miniaturized fluoroscope recorded mice swallowing an oral contrast agent [46].

Currently, pneumonia and aspiration pneumonia require new management targets. Nrf2 may be a candidate; however, its role in aspiration pneumonia, as well as in swallowing and respiratory muscles, is unclear.

This study aimed to identify the role of Nrf2 in the exacerbation of pneumonia, muscle atrophy, and swallowing dysfunction in aspiration pneumonia, as well as the role of Nrf2 in its exacerbation mechanism. Thus, we attempted to identify the unknown role of Nrf2 in aspiration pneumonia and the swallowing and respiratory muscles and explored the possibility of Nrf2 as a therapeutic target for the disease. To achieve this purpose, we caused aspiration pneumonia in wild-type and *Nrf2*-knockout mice and analyzed their swallowing and respiratory muscles.

## 2. Results

### 2.1. Induction of Severe Inflammation in Nrf2-Knockout Mice After Aspiration Challenge

Mice underwent an LPS and pepsin solution challenge 5 days per week to induce aspiration pneumonia, as shown previously [47]. Surprisingly, the *Nrf2*-knockout mice acutely lost their body weight and could not tolerate the challenge. Therefore, we reduced the challenge to 3 days a week. In addition, we reduced the amount of LPS solution during the first week from 20 to 10 μL. We evaluated the body weight change as a readout of an indirect whole-body effect of the challenge [8]. Wild-type mice did not lose body weight during the challenge until day 28 (Figure 1A). Conversely, compared with wild-types, *Nrf2*-knockout mice lost significantly more body weight from day 14 to day 28 (Figure 1A). Upon histological analysis, the control lungs of the wild-types and *Nrf2*-knockouts appeared indistinguishable (Figure 1B,E), even in higher-magnification images (Appendix A). Leukocytes infiltrated slightly, and lung edema was mild in challenged wild-types (Figure 1C). In challenged *Nrf2*-knockouts, leukocytes infiltrated massively, and lung edema was moderate to severe (Figure 1F). High-magnification images (Figure 1D,G) and the highest-magnification images (Appendix A) clearly showed the above characteristics (Figure 1D,G).

We evaluated inflammation by immunohistochemistry for the lung-infiltrated leukocytes. Controls of wild-types and *Nrf2*-knockouts revealed scarce infiltration of S100A8-immunoreactive neutrophils and F4/80-immunoreactive macrophages (Figure 1H,L). CD3e-immunoreactive T cells and B220-immunoreactive B cells revealed similar results in controls of wild-types and *Nrf2*-knockouts (Figure 1J,N). In challenged mice, large numbers of neutrophils and macrophages infiltrated in both types of mice (Figure 1I,M). We observed greater numbers of macrophages infiltrating in *Nrf2*-knockouts than in wild-types (Figure 1Q). A large number of B cells but not T cells infiltrated in the challenged wild-types and *Nrf2*-knockouts (Figure 1K,O). Greater numbers of B cells infiltrated in challenged *Nrf2*-knockouts than in wild-types (Figure 1S). The above data suggest that more severe inflammation was induced in the lungs of *Nrf2*-knockouts compared to wild-types by the aspiration challenge.

### 2.2. Induction of Proinflammatory Cytokines in Swallowing and Respiratory Muscles of Mice with Aspiration Challenge

We isolated the digastric muscles as the swallowing muscles to examine [48,49] and the diaphragms as the respiratory muscles to examine [9]. We chose IL-1β, IL-6, and MCP-1 as proinflammatory cytokines to examine, isolated mRNAs, and measured their expression levels on days 7, 14, 21, and 28 [47]. We set the expression levels of the controls to 100% and showed the relative expression levels of the challenged mice. When we performed Q-RT-PCR, the number of samples we could measure simultaneously was limited. Due to this technical limitation, we showed the mRNA levels of challenged mice compared to the controls of wild-types and *Nrf2*-knockouts, respectively. A comparison of control groups between wild-types and *Nrf2*-knockouts showed similar expression levels of the selected cytokines (Appendix A).

In the digastric muscles, challenged wild-types did not show elevated levels of *Il-1β* or *Mcp-1* compared to those on day 0. *Il-6* levels were elevated on days 21 and 28. In contrast, challenged *Nrf2*-knockouts showed elevated *Il-1β* and *Il-6* levels from day 14 and *Mcp-1* levels from day 21 to day 28. The levels of all cytokines in *Nrf2*-knockout mice were greatest on day 21 (Figure 2A). In the diaphragms, elevated levels of proinflammatory cytokines were detected in both groups of mice (Figure 2B). Wild-type mice showed elevated *Il-6* levels on day 28 and *Mcp-1* levels on day 21. *Nrf2*-knockout mice showed elevated *Il-1β* and *Mcp-1* levels from day 14 and elevated *Mcp-1* levels from day 7 to day 28. These results suggest that the period of elevated proinflammatory cytokine levels was longer in *Nrf2*-knockouts than in wild-types.

### 2.3. The Challenge Activated Calpains and Caspase-3 and Autophagy in the Swallowing and Respiratory Muscles

Proinflammatory cytokines induce the cleavage of myofibrillar proteins by activating calpain and caspase-3. We assessed their activation by evaluating fodrin using Western blotting. The activation of caspase-3 leads to the cleavage of fodrin into a 120 kDa fragment. Fodrin is cleaved into 145–150 kDa bands primarily by activated calpain, with a smaller contribution from caspase-3. In the digastric muscles of challenged wild-types and *Nrf2*-knockouts, the levels of fodrin immunoreactive bands did not change from those of controls (Figure 3A). These results suggest a low possibility of calpain and caspase-3 activation in the challenged digastric muscles. We evaluated the activation of autophagy by immunoreactive bands of P62, a protein associated with autophagy. Generally, low P62 levels suggest autophagy activation. The levels of P62 immunoreactive bands became lower than those of controls from day 7 to day 28 in the challenged digastric muscles of wild-types and *Nrf2*-knockouts (Figure 3A). The levels became the lowest on day 28, by 83% in wild-types and by 73% in *Nrf2*-knockouts.

In the diaphragms, the challenged wild-type mice showed elevated 120 and 145–150 kDa band levels on days 14 and 21 (Figure 3B). The challenged *Nrf2*-knockout mice showed elevated levels of both bands from day 7 to day 28 (Figure 3B). The levels of both bands were greatest on day 28 in *Nrf2*-knockout mice (Figure 3B). The above findings suggest calpain and caspase-3 activation in the challenged diaphragms of the wild-types and *Nrf2*-knockouts. The P62 band levels did not change between challenged and control mice in wild-types. In *Nrf2*-knockouts, challenged groups on days 14 and 28 showed lower p62 band levels than controls (Figure 3B).

The above findings suggest that in the digastric muscles, calpain and caspase-3 remained inactive, but autophagy was activated in wild-types and *Nrf2*-knockouts. In the diaphragms, calpain and caspase-3 activation occurred in wild-types and *Nrf2*-knockouts, but autophagy was activated only in *Nrf2*-knockouts.

### 2.4. The Ubiquitin–Proteasome Cascade Activation During the Challenge

After the cleavage, the myofibrils are broken down via the activated ubiquitin–proteasome cascade. The intensity of the ubiquitin–proteasome cascade activation was assessed by measuring the *Murf-1* and *Atrogin-1* mRNA levels. MuRF-1 and atrogin-1 are E3 ubiquitin ligases that are specific to skeletal muscles. A comparison of control mice revealed similar *Murf-1* expression levels between wild-types and *Nrf2*-knockouts (Appendix A). For *Atrogin-1*, *Nrf2*-knockouts exhibited greater expression than wild-types, with 3.4-fold greater expression in the digastric muscles and 1.6-fold greater expression in the diaphragms (Appendix A).

In the digastric muscles, the *Murf-1* and *Atrogin-1* mRNA levels were generally greater in the mice subjected to the challenge compared to the controls in wild-types and *Nrf2*-knockouts (Figure 4A). In the diaphragms, *Murf-1* levels were elevated on day 21 in wild-types and from day 7 to day 21 in *Nrf2*-knockouts. Atrogin-1 levels were elevated on day 14 in wild-types and from day 7 to day 28 in *Nrf2*-knockouts (Figure 4B). These findings suggest that the ubiquitin–proteasome system activation occurred within the challenged digastric muscles and diaphragms in wild-types and *Nrf2*-knockouts. In the diaphragms, the extended activation in *Nrf2*-knockouts suggests increased activity of the ubiquitin–proteasome system compared to that in wild-types.

### 2.5. Autophagy Activation in the Challenged Digastric Muscles and Diaphragms

We assessed the inclusion of autophagy by evaluating the expression levels of mRNA for genes associated with autophagy. The genes assessed were microtubule-associated protein 1 light chain 3B-II (*Lc3b*), Bcl2/adenovirus E1B 19 kDa interacting protein 3 (*Bnip3*), and GABA(A) receptor-associated protein like 1 (*Gabarapl1*) as shown previously [47]. A comparison of control mice revealed greater expression levels of the genes assessed in *Nrf2*-knockouts than in wild-types in both muscles (Appendix A). In the digastric muscles, expression levels were 3.5-fold (*Lc3b*), 2.0-fold (*Bnip3*), and 2.9-fold (*Gabarapl1*) greater in *Nrf2*-knockouts than in wild-types. In the diaphragms, they were 1.8-fold (*Lc3b*), 3.1-fold (*Bnip3*), and 1.4-fold (*Gabarapl1*) greater.

In the challenged groups, the wild-type digastric muscles showed elevated levels of *Lc3b* and *Bnip3* on day 28 and of *Gabarapl1* on days 14 and 21 (Figure 5A). In the *Nrf2*-knockout digastric muscles, *Lc3b* and *Bnip3* levels increased from day 7 to day 28. *Gabarapl1* levels increased from day 14 to 28 (Figure 5A). None of the assessed gene levels were elevated in the challenged wild-type diaphragms. The challenged *Nrf2*-knockout diaphragms showed elevated levels of *Lc3b* from day 7 to 28 and of *Bnip3* and *Gabarapl1* on days 7, 14, and 21 (Figure 5B).

The above findings suggest that autophagy activation in the digastric muscles was more extended in *Nrf2*-knockouts than in wild-types. In the challenged diaphragm, *Nrf2*-knockouts showed activation of the autophagy pathway, but wild-types did not.

### 2.6. Induction of Muscle Atrophy by Aspiration Challenge in Nrf2-Knockouts

We assessed the effects of the above molecular changes on muscle atrophy in the control group and the 28-day challenge group. The frequency distribution of fiber sizes was quantified by observing the cross-sectional dimensions of muscle tissues stained by hematoxylin and eosin. Observing the digastric muscles of controls, the distributions of fiber sizes were similar between the wild-types and *Nrf2*-knockouts (Figure 6A, the left panel). In the challenged digastric muscles, the muscle fibers shifted leftward in the *Nrf2*-knockouts but not in the wild-types, which suggests small fibers occupied a greater proportion in *Nrf2*-knockouts than in wild-types (Figure 6A, the right panel). The distributions of fiber sizes in the control diaphragms were similar between the wild-types and *Nrf2*-knockouts (Figure 6B, the right panel). In the challenged diaphragms, muscle fibers showed a more leftward shift in the *Nrf2*-knockouts than in the wild-types, which suggested greater small fiber proportions in the *Nrf2*-knockouts than in the wild-types (Figure 6B, the right panel).

The above findings suggest the distribution of similar myofiber sizes in control wild-types and *Nrf2*-knockouts in both muscles. The current study employed a lower-intensity aspiration challenge than did the previous study [47]. The low-intensity challenge induced greater atrophy in the digastric muscles and diaphragms in *Nrf2*-knockouts than in wild-types.

We compared the distributions of fiber sizes between control and challenged mice within wild-types and *Nrf2*-knockouts, respectively. In wild-types, the digastric muscles generally showed similar fiber size distributions in control and challenged mice. In contrast, muscle fibers shifted leftward in the challenged diaphragms (Appendix A). The muscle fibers shifted leftward in the challenged digastric muscles and diaphragms of *Nrf2*-knockouts (Appendix A).

These results suggest that the low-intensity challenge to the wild-types induced muscle atrophy in the diaphragms but not, or weakly, in the digastric muscles. In contrast, in the *Nrf2*-knockouts, the low-intensity challenge induced muscle atrophy in both muscles.

### 2.7. Effects of Muscle Atrophy on Swallowing Function

Swallowing muscle atrophy can impair swallowing function. Thus, we quantified the swallowing function of the challenged mice by a videofluoroscopic examination of swallowing. We kept the mice in a chamber equipped with an X-ray source and an X-ray camera (Figure 7A). Mice drank an oral contrast agent in a bowl, and their swallowing procedure was recorded (Figure 7B,C). We evaluated the bolus speed traversing the pharynx, mastication rate, and inter-swallow interval as indices of swallowing function. The mastication rate was the number of chews in one second. The inter-swallow interval was a time length between two successive sequential swallows. Control wild-types and *Nrf2*-knockouts and challenged wild-types showed no significant differences in these indices (Figure 7D–F). The above findings indicated that the control wild-types and *Nrf2*-knockouts and challenged wild-types had similar swallowing functions. Challenged *Nrf2*-knockout mice showed slower bolus speeds than the other groups (Figure 7D). The mastication rate was lower in the challenged *Nrf2*-knockouts than in the other groups (Figure 7E). The inter-swallow interval was significantly longer in the challenged *Nrf2*-knockouts than in the other groups (Figure 7F). These results suggest that swallowing function declined only in challenged *Nrf2*-knockouts.

## 3. Discussion

*Nrf2*-knockouts were more susceptible to the aspiration challenge than wild-types. This study demonstrated more exacerbated aspiration pneumonia, swallowing and respiratory muscle atrophy, and swallowing dysfunction in *Nrf2*-knockouts than in the wild-types, along with their underlying mechanisms. As the mechanisms, in the digastric muscles, the aspiration challenge induced more extended activation of the autophagy pathway in *Nrf2*-knockouts compared to wild-types. In the challenged diaphragms, the autophagy pathway was activated in *Nrf2*-knockouts but not in the wild-types. Moreover, the activation of the calpains/caspase-3 pathway was more extended in the challenged diaphragms of *Nrf2*-knockouts than in wild-types. We interpreted that the greater muscle atrophy of the swallowing and respiratory muscles in *Nrf2*-knockouts than in wild-types was induced by the greater activations of these muscle proteolysis pathways. The muscle atrophy resulted in a functional decline in swallowing in challenged *Nrf2*-knockouts.

The respiratory and appendicular muscles generate cytokines [50,51,52,53]. The cytokines and other peptides produced by muscles are named myokines [54,55,56]. The tongue produces myokines [47]. However, the ability of other swallowing muscles to produce myokines was not clear. In this study, the digastric muscles produced proinflammatory cytokines. These results confirmed the capacity of the swallowing muscles to produce myokines.

*Nrf2*-knockouts exhibited greater levels of proinflammatory cytokine expression than wild-types. A previous study reported that Nrf2 deficiency increased IL-1β production [57]. IL-1β induces the generation of IL-6 and MCP-1 [58]. We interpreted that these cytokine networks induced more severe inflammation in *Nrf2*-knockouts than in wild-types.

In the diaphragms, *Nrf2*-knockouts showed extended activation of the calpains/caspase-3 pathway compared to wild-types. In addition, autophagy pathways were activated in *Nrf2*-knockouts but not in wild-types. This was consistent with previous data; autophagy had not been activated in the diaphragms of a wild-type aspiration pneumonia mouse model [47]. These activations of muscle proteolysis pathways may induce greater diaphragm atrophy in *Nrf2*-knockouts than in wild-types.

The expression level of *Atrogin1* in the unstimulated digastric muscles and diaphragms of *Nrf2*-knockouts was enhanced. Generally, NF-κB regulates *Atrogin1* expression. In muscles of *Nrf2*-knockouts, reactive oxygen species accumulated even without stimulation [36,59]. Reactive oxygen species activate NF-κB [57,60]. Therefore, we speculated that accumulated reactive oxygen species due to Nrf2 deficiency might activate NF-κB and increase the expression of *Atrogin1*.

Furthermore, the challenge activated the ubiquitin–proteasome cascade in the digastric muscles and diaphragms. However, since caspase-3 and calpains, the upstream muscle degradation pathways of the ubiquitin–proteasome cascade, were not activated in the digastric muscles, their possible involvement in muscle atrophy was interpreted as low. The involvement of the ubiquitin–proteasome cascade in “Nrf2 activation” is widely recognized [61,62]. However, it is unclear whether Nrf2 deficiency directly affects the triggering of the ubiquitin–proteasome cascade. Instead, we speculated that the exacerbation of inflammation due to Nrf2 deficiency might have activated the ubiquitin–proteasome pathway [11].

Previous studies observing lungs showed essential roles of Nrf2 in lung protection from various inflammatory stimulations, such as butylated hydroxytoluene, carrageenin, LPS, and TNF-α [32,41,63]. *Nrf2*-knockout mice showed greater lung inflammation and a lower survival rate than wild-types [41,63]. These studies showed the effects of Nrf2 in acute lung inflammation. The current study employed chronic lung inflammation and showed greater inflammatory response and muscle atrophy with a lower survival rate in a preliminary experiment in *Nrf2*-knockout mice than in wild-types. Thus, Nrf2 is essential in lung protection in acute and chronic inflammation.

The challenge induced an extended mRNA elevation of autophagy-related genes in *Nrf2*-knockouts compared to those in wild-types. This difference was detected in both the digastric muscles and the diaphragms, suggesting greater autophagy activation in *Nrf2*-knockouts. Similarly, a previous report showed that autophagy was enhanced in the skeletal muscles of *Nrf2*-knockouts [36]. As for its mechanism, previous studies suggested reactive oxygen species accumulated in the *Nr2*-knockout muscles [36,59]. Indeed, reactive oxygen species are produced by proinflammatory cytokines in cells [64]. Reactive oxygen species accumulation within muscles generally enhances autophagy and is linked to the muscle proteolysis pathway [12,65,66]. Consistent with these results, histological examination revealed that muscle atrophy in the digastric muscles and diaphragms was enhanced in *Nrf2*-knockouts compared to wild-types. Thus, Nrf2 may suppress muscle atrophy through the autophagy regulation. However, in the atrophy of the wild-type diaphragms, the possibility of autophagy involvement was considered low based on the mRNA and p62 protein levels.

Nrf2 regulates p62 expression; however, its protein levels were not different between wild-types and *Nrf2*-knockouts. This might be because other transcription factors, such as NF-κB, also control the p62 expression, and their role was strong in muscles [67].

Differences in body weights between wild-types and *Nrf2*-knockouts were observed from day 14 after the challenge. Although the degree of body weight recovery after day 20 may reflect poor oral intake due to pneumonia, there was still a difference, suggesting the involvement of muscle atrophy.

Aspiration pneumonia causes chronic inflammation induced by repeated micro-aspiration, such as recurrence of saliva aspiration. This type of aspiration syndrome is known as diffuse aspiration bronchiolitis [47]. The three-time-a-week aspiration challenge in this study may induce diffuse aspiration bronchiolitis and pneumonia.

The limitations of this study are as follows. In quantitative RT-PCR, the number of samples that could be detected simultaneously was limited to 96 wells per plate, and we measured each sample in duplicate, which posed a technical limitation. As a result, we could not directly compare the mRNA levels of *Nrf2*-knockouts and wild-types. We showed mRNA levels of challenged mice in comparison to the controls of wild-types and *Nrf2*-knockouts, respectively. The current study suggests the protective role of Nrf2 in lung inflammation. However, we could not overexpress the Nrf2 and confirm its anti-inflammatory effects in this study.

The challenge generated more severe pneumonia and muscle atrophy in *Nrf2*-knockouts than in wild-types. The above findings suggested that Nrf2 alleviated inflammation and muscle atrophy in pneumonia. The swallowing and respiratory muscle atrophy weakens one’s capacity to swallow and cough and may induce the recurrence of pneumonia. Thus, activation of Nrf2 could effectively limit the development of pneumonia and prevent its recurrence.

## 4. Materials and Methods

### 4.1. Mice

Female C57BL/6J background *Nrf2*-knockout mice aged 7 to 10 weeks and their littermate controls (wild-types) were kept under barrier conditions [68]. Generally, under barrier conditions without various stresses, the growth rate and fertility of *Nrf2*-knockout mice are similar to those of their littermate controls [15]. We injected ketamine and xylazine intraperitoneally and anesthetized mice [8]. Prof. Hozumi Motohashi kindly provided the mice that were bred and maintained at a Tohoku University animal facility. We obtained approval from The Tohoku University Laboratory Animal Committee for the entire procedure.

### 4.2. Developing an Aspiration Pneumonia Model Using Mice

We developed aspiration pneumonia in mice, as demonstrated in a prior study [8]. To mimic food, we added toromerin^®^ (Sanwa Kagaku Kenkyusho, Nagoya, Japan) to PBS, thickening it to a concentration of 12 mg/mL. To mimic gastric juice, we adjusted PBS to pH 1.6 by HCl and dissolved pepsin (2 mg/mL; Sigma, St. Louis, MO, USA). LPS (Sigma) was suspended in PBS at 2.5 mg/mL. Mice were anesthetized and intranasally inoculated with 25 μL of pepsin suspended in PBS at 2 mg/mL, and 10 or 20 μL of LPS suspended in PBS at 2.5 mg/mL, respectively, as a challenge, 3 times per week for the indicated days.

### 4.3. Extraction of RNA and Quantitative Real-Time Polymerase Chain Reaction (Q-RT-PCR) Protocol

We conducted RNA extraction and Q-RT-PCR according to former studies [9,47]. We isolated the total muscle RNA employing RNAzol (Molecular Research Center Inc., Cincinnati, OH, USA). Then, we cleaned it using the ReverTra Ace qPCR RT Master Mix (Toyobo Co., Osaka, Japan). We designed primers for the specific genes and measured their expression levels, as demonstrated previously (Appendix A) [9]. Primer dimer formation or contamination was evaluated by analyzing the dissociation curve of each PCR assay. We chose *β-actin* or *Gapdh* as the housekeeping gene. The expression levels of mRNA were assessed by determining the fold change compared to the control group, and their relative expression degrees were examined. We ran Q-RT-PCR applying THUNDERBIRD SYBR qPCR Mix (Toyobo Co., Osaka, Japan) and THUNDERBIRD Probe qPCR Mix (Toyobo Co.) with the StepOne Plus Real-Time PCR System (Applied Biosystems, Foster City, CA, USA).

### 4.4. Western Blotting Process

We conducted Western blotting as demonstrated in former studies [9,47]. The following are the antibody dilutions and their manufacturers (Appendix A): anti-α-fodrin (1:1000; ENZO, New York, NY, USA), anti-p62 (1:1000; MBL, Tokyo, Japan), anti-α-Tubulin (1:2000; Sigma-Aldrich, Saint Louis, MO, USA), HRP-conjugated Goat Anti-Mouse IgG (H + L) (1:2000; Proteintech, Rosemont, IL, USA), and HRP-conjugated Goat Anti-Guinea pig IgG (H + L) (1:2000; Proteintech). Chemi-Lumi One L (Nacalai Tesque, Kyoto, Japan) was used to visualize the bands, and the Chemi Doc MP Imaging System (BIO RAD, Hercules, CA, USA) was used to quantify their intensities. 

### 4.5. Histological Analysis and Immunohistochemistry

We perfused the mice, isolated the digastric muscles, diaphragms, and lungs, stored them at −80 °C, and prepared cryosections as previously shown [8,47]. Cryostat sections with a thickness of 10 μm (muscles) or 7 μm (lungs) were stained using hematoxylin and eosin or primary antibodies dissolved in PBS that contained 0.3% Triton X-100, 0.2% bovine serum albumin, 5% normal goat serum, and 0.1% sodium azide [8]. The dilution and antibodies used primarily for immunohistochemistry were as follows (Appendix A): neutrophils: S100 calcium-binding protein A8 (S100A8; 1:500; goat; AF3059, R&D Systems, Minneapolis, MN, USA); macrophages: F4/80 (1:100; Rat; MCA497G, Bio-Rad, UK); T cells: CD3e (1:500; hamster; eBio500A2, eBioscience, San Diego, CA, USA); B cells: B220/CD45R (1:500; rat; RA3-6B2, eBioscience). Cy3- (Jackson ImmunoResearch Laboratories Inc., West Grove, PA, USA) or FITC-labeled species-specific secondary antibodies were applied. We obtained muscle images utilizing a BZ-9000 microscope (Keyence, Osaka, Japan) and evaluated them employing BZ-II (ver 10.3.2, Keyence) software. We analyzed muscle tissue sections by measuring 500 muscle fibers per sample. We took fluorescence images and evaluated the infiltration of neutrophils, macrophages, T cells, and B cells by measuring their area densities. We took 10 images for each sample using an objective lens (×20). We measured area densities as the percentage of pixels with fluorescence intensity at or above the empirically determined threshold levels [69]. We obtained fluorescence images with a confocal microscope, LSM 800 (Carl Zeiss, Jena, Germany), equipped with Zen (Blue 2.3) software [69].

### 4.6. Videofluoroscopic Examination of Swallowing

Each mouse underwent a videofluoroscopic examination of swallowing 2 times before and after the 28-day challenge. Before each examination, mice were restricted for fluid overnight. During the examination, each mouse was kept in a customized chamber with a small fluoroscope (The Labscope, Glenbrook Technologies, Randolph, NJ, USA). Lab Scope video software (GTI-2000 Real-Time Image Workstation Ver1.30.0, Glenbrook Technologies) recorded the swallowing procedure for 30 to 60 s as audio video interleave files (30 frames/s). A contrast solution administered orally (Omnipaque, GE Healthcare Japan Inc., Tokyo, Japan, 350 mg iodine/mL) was thinned to a 50% concentration using water mixed with chocolate syrup for flavoring. The viscosity of the solution was adjusted to 0.0039 Pa·s, as measured by a tuning fork vibro viscometer (A&D, Inc., Tokyo, Japan). The agent was kept in a bowl on the chamber floor. Mice drank an oral contrast agent. We quantified swallowing impairment in mice by analyzing the following indices as shown previously: bolus speed through the pharynx (mm/s), mastication rate (cycles/s), and inter-swallow interval (second) [44,70]. The bolus speed through the pharynx was measured as the speed from the vallecula to the 2nd cervical vertebra. The mastication rate was defined as the number of cycles from mandible opening to closing within one second. The inter-swallow interval refers to the duration between two sequential successive swallows. The recorded videos were analyzed frame by frame.

### 4.7. Statistical Analysis

We showed values as mean values ± standard errors. Body weights were compared at each time point, and fiber sizes were compared at the same fiber size between wild-types and *Nrf2*-knockouts using a two-tailed paired Student’s *t*-test. For other values, we statistically analyzed the data using one-way analysis of variance with Tukey’s multiple comparison test as a post hoc. We employed JMP Pro v. 16.0 software (SAS Institute Inc., Cary, NC, USA) for the data analysis. We interpreted *p* values below 0.05 to have significant differences.

## 5. Conclusions

This research assessed the involvement of Nrf2 in aspiration pneumonia, followed by atrophy and dysfunction of aspiration-related muscles, in a preclinical animal model. The new findings in this study suggest that Nrf2 is important for alleviating severe pneumonia, protecting muscles from atrophy, and maintaining muscle function. Since aspiration pneumonia is widespread and fatal in an aging society, Nrf2 may have the potential to become a novel target for protecting patients from serious health conditions. Indeed, there is a worldwide competition to develop Nrf2 activators, and some are currently approved for Friedreich ataxia and multiple sclerosis [71,72]. A future perspective may involve clinical trials on Nrf2 activators in aspiration pneumonia.

## Figures and Tables

**Figure 1 ijms-25-11829-f001:**
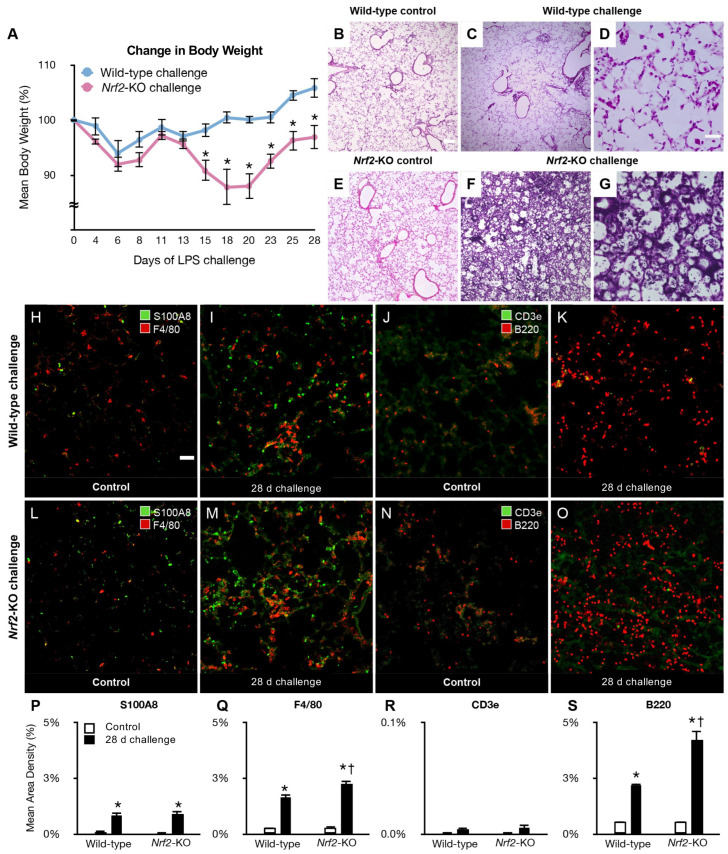
(**A**–**G**): The changes in body weight and lung pathology in challenged mice. (**A**): The body weight change during the 28-day aspiration challenge. * *p* < 0.05 vs. wild-types. (**B**–**G**)**:** The lung tissue sections that are representative of controls ((**B**): wild-types, (**E**): *Nrf2*-knockouts), and after 28 days of aspiration challenge ((**C**,**D**): wild-types, (**F**,**G**): *Nrf2*-knockouts) stained for hematoxylin and eosin. (**H**–**O**): Confocal images of leucocytes infiltrating into the lungs in wild-type (**H**–**K**) or *Nrf2*-knockout (**L**–**O**) mice after a 28-day challenge (**I**,**K**,**M**,**O**) or in controls (**H**,**J**,**L**,**N**). A few S100A8-immunoreactive neutrophils (green) and F4/80-immunoreactive macrophages (red) infiltrated the controls (**H**,**L**). Many neutrophils and macrophages infiltrated the challenged lungs (**I**,**M**). We recognized a similar trend in the infiltrations of CD3e-immunoreactive T cells (green) and B220-immunoreactive B cells (red) (**J**,**K**,**N**,**O**). (**P**–**S**): The area densities of infiltrated leukocytes were differentially quantified. Macrophages and B cells infiltrated more in *Nrf2*-knockouts than in wild-types in challenged groups (**Q**,**S**). Scale bars: (**B**,**C**,**E**,**F**): 500 µm in (**B**). (**D**,**G**): 200 µm in (**D**). (**H**–**O**): 50 µm in (**H**). * *p* below 0.05 versus controls, † *p* below 0.05 versus challenged wild-types. Five mice per group; data are from 2 individual experiments.

**Figure 2 ijms-25-11829-f002:**
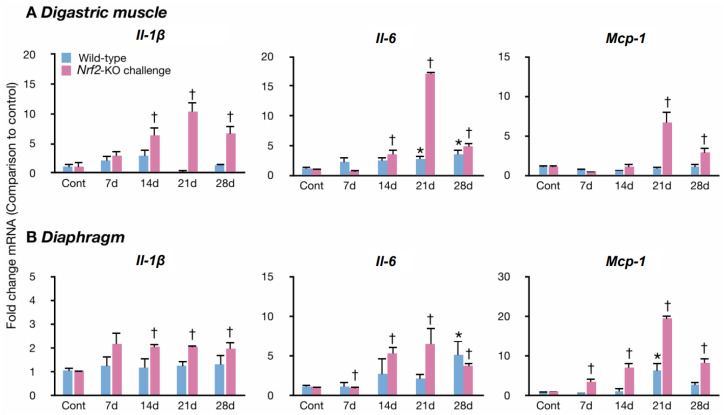
The proinflammatory cytokine mRNA expression levels in the challenged muscles. (**A**,**B**): Q-RT-PCR determined *Il-1β* (left panels), *Il-6* (center panels), and *Mcp-1* (right panels) mRNA levels as proinflammatory mediators in the digastric muscles (**A**) and the diaphragms (**B**). Cont, control mice; 7 d, mice challenged for 7 days; 14 d, mice challenged for 14 days; 21 d, mice challenged for 21 days; 28 d, mice challenged for 28 days. Indicated levels are fold changes in comparison to the average values of control wild-types or *Nrf2*-knockouts, respectively; * *p* below 0.05 versus controls of wild-types. † *p* below 0.05 versus controls of *Nrf2*-knockouts. Five mice per group, representative of 2–3 individual trials.

**Figure 3 ijms-25-11829-f003:**
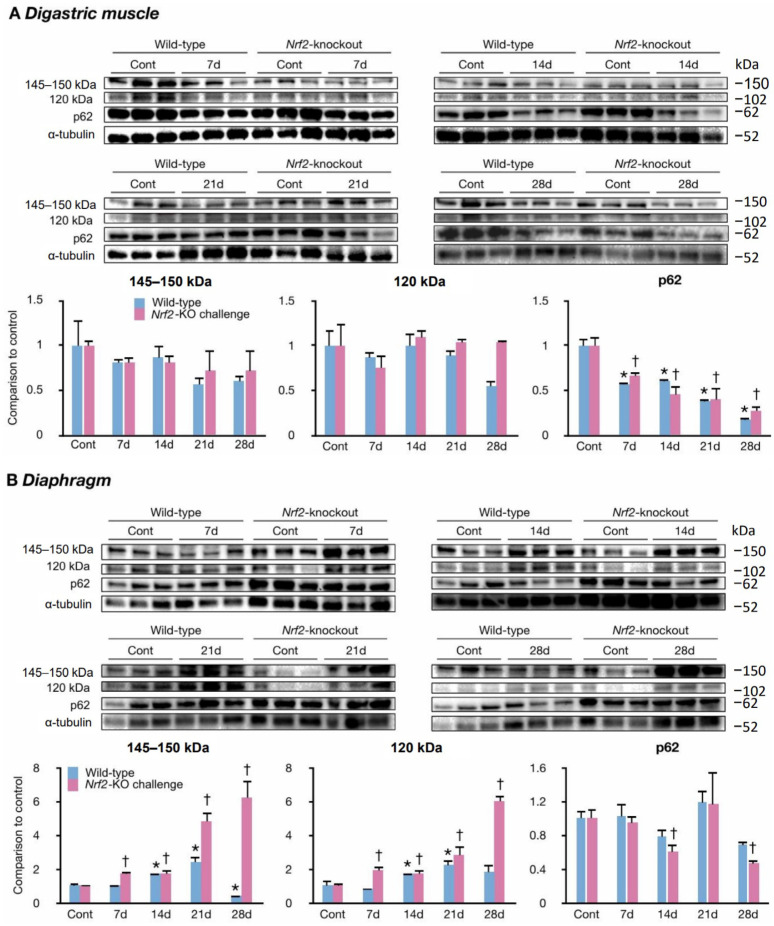
The activations of the proteases in the challenged muscles. (**A**,**B**): Western blot analysis and quantification of bands immunoreactive with fodrin or p61. Caspase-3 cleavage products are at 120 kDa, and mostly calpain-mediated cleavage products are at 145–150 kDa and 62 kDa (p62) in the digastric muscles (**A**) and the diaphragms (**B**). Cont, control mice; 7 d, mice challenged for 7 days; 14 d, mice challenged for 14 days; 21 d, mice challenged for 21 days; 28 d, mice challenged for 28 days. Indicated levels are fold changes in comparison to the average values of control wild-types or *Nrf2*-knockouts, respectively; * *p* below 0.05 versus controls of wild-types. † *p* below 0.05 versus controls of *Nrf2*-knockouts. Three mice in each group; data are from 2–3 individual experiments.

**Figure 4 ijms-25-11829-f004:**
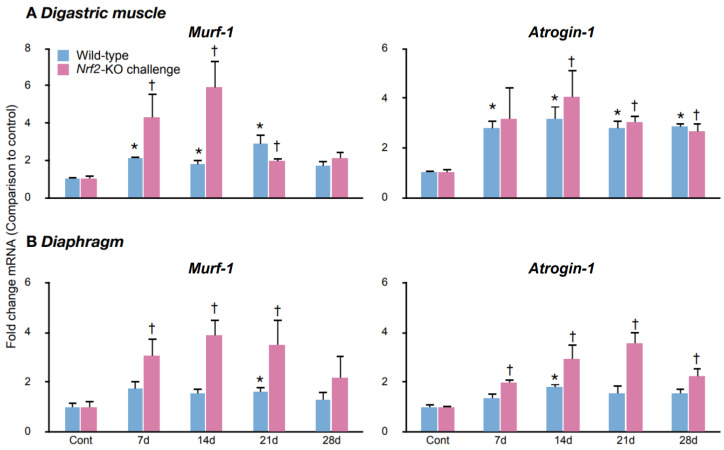
The temporal profiles of *Murf-1* and *Atrogin-1* expression. (**A**,**B**): Q-RT-PCR evaluated the mRNA expression degrees of the E3 ubiquitin ligases, *Murf-1* (left panels) and *Atrogin-1* (right panels), which are muscle-specific, in the digastric muscles (**A**) and the diaphragms (**B**). Cont, control mice; 7 d, mice challenged for 7 days; 14 d, mice challenged for 14 days; 21 d, mice challenged for 21 days; 28 d, mice challenged for 28 days. Indicated levels are fold changes in comparison to the average values of control wild-types or *Nrf2*-knockouts, respectively; * *p* below 0.05 versus controls of wild-types. † *p* below 0.05 versus controls of *Nrf2*-knockouts. Five mice per group, representative of 2–3 individual trials.

**Figure 5 ijms-25-11829-f005:**
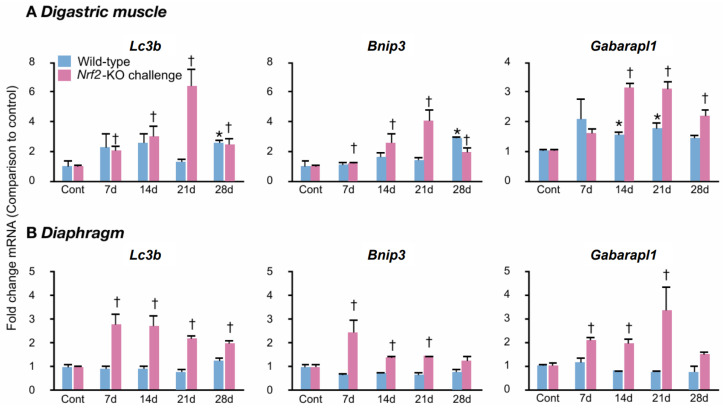
Autophagy-associated gene activation in the challenged muscles. (**A**,**B**): Q-RT-PCR determined the mRNA levels of autophagy-involved genes, *Lc3b* (left panels), *Bnip3* (center panels), and *Gabarapl1* (right panels), in the digastric muscles (**A**) and the diaphragms (**B**). Cont, control mice; 7 d, mice challenged for 7 days; 14 d, mice challenged for 14 days; 21 d, mice challenged for 21 days; 28 d, mice challenged for 28 days. Indicated levels are fold changes in comparison to the average values of control wild-types or *Nrf2*-knockouts, respectively; * *p* below 0.05 versus controls of wild-types. † *p* below 0.05 versus controls of *Nrf2*-knockouts. Five mice per group, representative of 2–3 individual trials.

**Figure 6 ijms-25-11829-f006:**
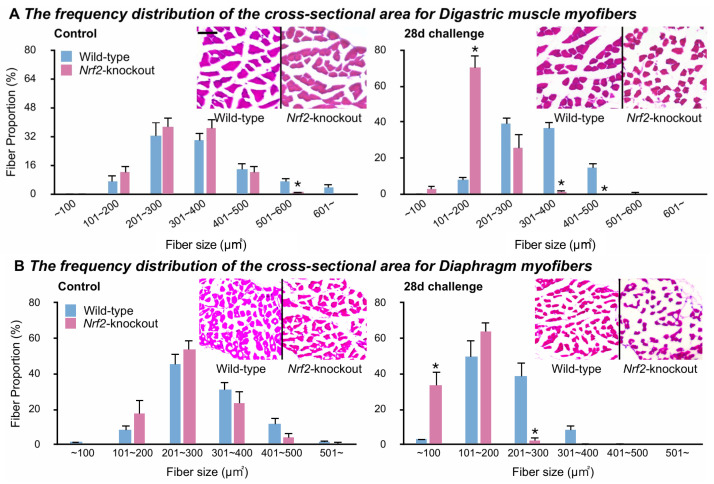
Muscle atrophy induced by the aspiration challenge in *Nrf2*-knockout mice. (**A**,**B**): The digastric muscles (**A**) and diaphragms (**B**) were isolated from wild-types and *Nrf2*-knockouts. The left panels show data from control mice, and the right panels show data from 28-day challenged mice. The graphs of frequency distributions of fiber sizes and the representative images of each group are shown. * *p* less than 0.05 vs. wild-types at the same fiber size. Five mice per group, representative of 2 individual trials. The scale bar in the left upper panel applies to all the images: 50 µm.

**Figure 7 ijms-25-11829-f007:**
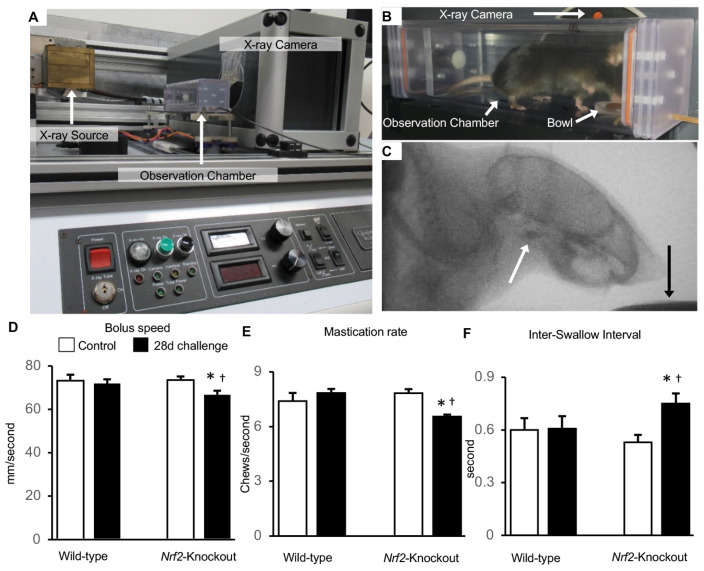
The swallowing muscle atrophy resulted in impaired swallowing function. (**A**–**C**): The swallowing function was videofluoroscopically examined using the LabScope. The LabScope equips an X-ray source and an X-ray camera with an observation chamber (**A**). A mouse in the chamber drank an oral contrast agent in a bowl (**B**). The fluoroscopy shows the entire head and cranial side of the thorax of a mouse. The white arrow shows the contrast agent in the oropharynx, and the black arrow shows the contrast agent in the bowl (**C**). (**D**–**F**): The speed from the vallecula to the 2nd cervical vertebra is shown as the bolus speed (**D**). The number of cycles from the mandible opening to closing in one second is shown as the mastication rate (**E**). The time length between 2 sequential successive swallows is shown as the inter-swallow interval (**F**). * *p* below 0.05 versus wild-types. † *p* below 0.05 versus *Nrf2*-knockout controls. Five mice per group.

## Data Availability

The data sets are available from the corresponding author upon reasonable request.

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
