# Peer review of "Nrf2 Deficiency Exacerbates the Decline in Swallowing and Respiratory Muscle Mass and Function in Mice with Aspiration Pneumonia"

_ijms, 2024, doi:10.3390/ijms252111829_

Round 1
Reviewer 1 Report
Comments and Suggestions for Authors
In this manuscript authors explored the role of Nrf2 in aspiration pneumonia by examining the muscle mass and function of swallowing and respiration using Nrf2-knockout mice. Authors found that Nrf2-knockouts exacerbated aspiration pneumonia and exhibited more persistent and intense proinflammatory cytokine elevation than wild-types. In both mice, the challenge activated calpains and caspase-3 in the diaphragm but not in the digastric muscles. The digastric muscles showed extended autophagy activation in Nrf2-knockouts compared to wild-types. The diaphragms exhibited autophagy activation only in Nrf2-knockouts. Moreover, Nrf2-knockouts showed worsened both muscle atrophies and swallowing function than wild-types.
manuscript and topic are very interesting. Figures are quite clear but several points deserve to be improved. In particular;
Lines 57-64: Since NRF2/KEAP1 signaling plays a key role in this manuscript, the multifaceted role of this pathway deserves to be highlighted. In fact, this signaling plays key functions in the onset and progression of several diseases including cancer (see PMID: 37296999 ).
Figure 1: Higher magnifications are needed to appreciate tissue morphology and staining
Figure 3: Molecular weights must be added. Correct "a-tubrin" with a-tubulin
Figure 7A-C are too small
4.4. Western blotting process: Authors must add the product code of all primary antibodies used
I suggest to move the primary antibodies used for WB, IF and IHC in a dedicate table
An accurate revision of syntax and terminology is necessary.
Author Response
添付ファイルをご覧ください。

Reviewer 2 Report
Comments and Suggestions for Authors
The topic is interesting and the paper is quite well written. I have some comments:
1) Aspiration pneumonia exacerbates swallowing and respiratory muscle atrophy. It induces respiratory muscle atrophy through three steps: proinflammatory cytokine production, caspase-3 and calpain, then ubiquitin-proteasome activations. Meanwhile, swallowing muscle atrophy is induced by autophagy. Nrf2 is the central detoxifying and antioxidant gene whose function in aspiration pneumonia is unclear. We explored the role of Nrf2 in aspiration pneumonia by examining the muscle mass and function of swallowing and respiration using Nrf2-knockout mice. Pepsin and lipopolysaccharide aspiration challenges to wild-type and Nrf2-knockout mice caused aspiration pneumonia. The swallowing (digastric muscles) and respiratory (diaphragm) muscles were isolated. Quantitative RT-PCR and western blotting assessed their proteolysis cascade. Pathological and videofluoroscopic examinations evaluated their atrophy and swallowing function, respectively. Nrf2-knockouts showed exacerbated aspiration pneumonia than wild-types. Nrf2-knockouts exhibited more persistent and intense proinflammatory cytokine elevation than wild-types. In both mice, the challenge activated calpains and caspase-3 in the diaphragm but not in the digastric muscles. The digastric muscles showed extended autophagy activation in Nrf2-knockouts compared to wild-types. The diaphragms exhibited autophagy activation only in Nrf2-knockouts. Nrf2-knockouts showed worsened both muscle atrophies and swallowing function than wild-types. Thus, Nrf2 may become a therapeutic target for aspiration pneumonia, an aging population’s major health problem. Abstract might be beneficial to include a sentence that briefly summarizes the key findings of the study. This can provide readers with a quick overview of the research.
2) 1. Introduction 33 Among the chief causes of mortality, the combination of pneumonia and aspiration 34 pneumonia occupied the third or fourth in Japan [1]. A recent study reported that aspi- 35 ration pneumonia occupied 42% of pneumonia inpatients aged >70 years [2]. Impaired 36 reflexes of swallowing and coughing are two critical factors that increase the risk of aspi- 37 ration pneumonia [3]. I suggest to improve the introduction.
3) We hypothesized that Nrf2 inactivation would induce severe pneumonia, muscle at- 79 rophy, and a decline in swallowing capacity. We aimed to identify the function of Nrf2 80 in aspiration pneumonia and muscle atrophy and its possibility for managing aspiration 81 pneumonia. To achieve this purpose, we caused aspiration pneumonia in wild-type and 82 Nrf2-knockout mice and analyzed their swallowing and respiratory muscles. Please, improve the the description of study aim and underline the novelty of the study.
4) 2. Results. Please, underline the most important results to clarify the conclusions.
5) 3. Discussion 315 Nrf2-knockouts were more susceptible to the aspiration challenge than wild-types. 316 In the digastric muscles, the aspiration challenge induced more extended activation of the 317 autophagy pathway in Nrf2-knockouts compared to wild-types. In the challenged dia- 318 phragms, the autophagy pathway was activated in Nrf2-knockouts but not in the wild- 319 types. The discussion section needs to be improved. It is necessary to clarify the results obtained and compare them with previous or similar studies.
6) The challenge generated more severe pneumonia and muscle atrophy in Nrf2-knock- 388 outs than in wild-types. The above findings suggested that Nrf2 alleviated inflammation 389 and muscle atrophy in pneumonia. The swallowing and respiratory muscle atrophy 390 weakens one’s capacity to swallow and cough and may induce the recurrence of pneumo- 391 nia. Thus, activation of Nrf2 could effectively limit the development of pneumonia and 392 prevent its recurrence. I suggest to underline the limitations of the study.
7) Conclusions. Please, underline the novelty of the study.
8) 4.7. Statistical analysis 473 We showed values as mean values ± SEMs. We statistically analyzed the data using 474 ANOVA with Tukey’s multiple comparison test. We interpreted P values below 0.05 to 475 have significant differences. Improve this paragraph and ameliorate the description of statistical tests used to evaluate the data
Author Response
有益で独創的なコメントをありがとうございました。あなたのコメントに従って改善を行った結果、原稿は強化されブラッシュアップされ、読者の興味を強く引くものとなりました。
レビュー担当者へのポイントツーポイントの返信
査読者 1 への回答
57-64行目: NRF2/KEAP1シグナル伝達は本稿で重要な役割を果たしているため、この経路の多面的な役割は強調する価値があります。実際、このシグナル伝達は、がんを含むいくつかの疾患の発症と進行において重要な役割を果たします(PMID: 37296999を参照)。
レビュー担当者のコメントに対する回答:
貴重なご意見ありがとうございます。NRF2の多面的な役割を強調するため、以下の文章を追加しました。
2ページ、61-62行目
核因子赤血球2関連因子2(Nrf2)は、様々な遺伝子発現を制御し、酸化ストレスへの応答と防御を調節する[14-17]。
2ページ、67-72行目
その後、Nrf2は安定化して核に移動し、細胞の分化、保護、増殖に関連する多数の遺伝子の活性化を開始します[21-24]。したがって、Nrf2は癌において重要な役割を果たしており[25-27]、特に化学療法や放射線療法に対する感受性の標的として重要な役割を果たしています[28-30]。さらに、Nrf2は肺を含むさまざまな場所で強力な抗炎症作用を発揮します[31-33]。
2ページ、76-77行目
さらに、Nrf2は抗老化プロセスにも寄与する[42]。
図 1 : 組織の形態と染色を評価するには、より高い倍率が必要です。
レビュー担当者のコメントに対する回答:
おっしゃる通りです。補足図 1 に肺の拡大画像を追加しました。
図3 : 分子量を追加する必要があります。「a-tubrin」を「a-tubulin」に修正します。
レビュー担当者のコメントに対する回答:
ご丁寧なアドバイスをありがとうございます。以前に示したとおり分子量を追加しました(参考文献 #53)。ご指摘のとおりスペルを修正しました。
図7A-Cは小さすぎる
レビュー担当者のコメントに対する回答:
はい、おっしゃる通りです。読者の皆様に分かりやすくするために、写真を撮り直し、大きめの画像に変更しました。
4.4. ウェスタンブロッティングプロセス:著者は使用したすべての一次抗体の製品コードを追加する必要がある。
WB、IF、IHCに使用する一次抗体を専用のテーブルに移動することを提案します。
レビュー担当者のコメントに対する回答:
ご提案ありがとうございます。ご指示に従い、使用した一次抗体を補足表 2 にまとめました。
構文と用語の正確な改訂が必要です。
詳細なコメントをありがとうございます。遺伝子名を修正しました。遺伝子の最初の文字を大文字にし、原稿全体で遺伝子を斜体にしました。微小管関連タンパク質 1 軽鎖 3B-II (LC3B-II) の略語をLc3bに変更しました(7 ページ、2443 行目)。これらの修正により、原稿が大幅に改善されました。改めて感謝申し上げます。
査読者2への回答
- 要約には、研究の主要な調査結果を簡潔にまとめた文章を含めるとよいでしょう。これにより、読者は研究の概要をすぐに把握できます。
レビュー担当者のコメントに対する回答:
貴重なアドバイスをいただき、誠にありがとうございます。下記の通り概要に追加させていただきました。
1ページ、33-34行目;
したがって、Nrf2 の活性化は、高齢者の主な健康問題である誤嚥性肺炎における炎症、筋萎縮、機能低下を軽減する可能性があり、治療の標的となる可能性があります。
- 紹介文を改善することを提案します。
レビュー担当者のコメントに対する回答:
貴重なご意見をありがとうございます。あなたのアドバイスと他の査読者のコメントに基づいて、序文に大幅な改訂を加えました。
- 研究目的の説明を改善し、研究の新規性を強調してください。
レビュー担当者のコメントに対する回答:
原稿を改善するための素晴らしいご提案をありがとうございます。以下のように序文に追加させていただきました。
2ページ、87-91行目
本研究では、誤嚥性肺炎における肺炎の増悪、筋萎縮、嚥下障害、および増悪メカニズムにおけるNrf2の役割を明らかにすることを目的とし、誤嚥性肺炎および嚥下・呼吸筋におけるNrf2の未知の役割を明らかにし、Nrf2が本疾患の治療標的となる可能性を探りました。
- 結論を明確にするために、最も重要な結果に下線を引いてください。
レビュー担当者のコメントに対する回答:
アドバイスありがとうございます。最も重要な結果を、以下のように考察セクションの最初の段落に書きました。
10ページ、343-345行目
この研究では、野生型よりもNrf2ノックアウトで誤嚥性肺炎、嚥下筋および呼吸筋の萎縮、嚥下機能障害がより悪化していることが示され、その根本的なメカニズムも示されました。
- 考察セクションは改善が必要です。得られた結果を明確にし、以前の研究や類似の研究と比較する必要があります。
レビュー担当者のコメントに対する回答:
貴重なアドバイスありがとうございます。以下の文章を追記させていただきました。
11ページ、387-394行目
肺を観察したこれまでの研究では、ブチルヒドロキシトルエン、カラギーナン、LPS、TNF-αなどのさまざまな炎症刺激からの肺の保護にNrf2が重要な役割を果たすことが示されています[32,41,61]。Nrf2ノックアウトマウスは、野生型よりも肺の炎症が大きく、生存率が低かった[41,61]。これらの研究は、急性肺炎症におけるNrf2の効果を示しています。現在の研究では慢性肺炎症を採用し、予備実験では、野生型よりもNrf2ノックアウトマウスで炎症反応が大きく、筋萎縮が見られ、生存率が低かった。このように、Nrf2は急性および慢性炎症における肺の保護に不可欠です。
- 研究の限界を強調することを提案します。
レビュー担当者のコメントに対する回答:
ご指摘ありがとうございます。以下の説明を追加しました。
12ページ、420-427行目
この研究の限界。定量的 RT-PCR では、同時に検出できるサンプルの数はプレートあたり 96 ウェルに制限されており、各サンプルを 2 回測定するという技術的な制限がありました。その結果、Nrf2ノックアウトと野生型の mRNA レベルを直接比較することはできませんでした。野生型とNrf2ノックアウトの対照群と比較して、チャレンジしたマウスの mRNA レベルを示しました。現在の研究は、肺の炎症における Nrf-2 の保護的役割を示唆しています。ただし、この研究では Nrf2 を過剰発現させてその抗炎症効果を確認することはできませんでした。
- この研究の斬新さを強調してください。
レビュー担当者のコメントに対する回答:
素晴らしいご提案をありがとうございます。新規性は以下のように強調されました。
14ページ、525-526行目
この研究の新たな発見は、Nrf2 が重度の肺炎を緩和し、筋肉の萎縮を防ぎ、筋肉の機能を維持するために重要であることを示唆しています。
- この段落を改善し、データの評価に使用される統計テストの説明を改善します。
レビュー担当者のコメントに対する回答:
方法についての情報が不十分であり、大変申し訳ございませんでした。説明を追加し、以下のとおり方法を改善しました。
13ページ、519-525行目
値は平均値±標準誤差として示しました。体重は各時点で比較し、繊維サイズは同じ繊維サイズで野生型とNrf2ノックアウト間で両側対応のあるスチューデントt検定を使用して比較しました。他の値については、事後検定としてTukeyの多重比較検定を伴う一元配置分散分析を使用してデータを統計的に分析しました。データ分析にはJMP Pro v. 16.0ソフトウェア(SAS Institute Inc.、ノースカロライナ州ケアリー)を使用しました。
査読者3への回答
4 ページ、130 ~ 132 行目: 「嚥下筋」、「横隔膜」、「炎症性サイトカイン」の後に「検査する」を追加してください。
レビュー担当者のコメントに対する回答:
詳細なコメントをいただきありがとうございます。以下のように修正させていただきました。
4ページ、141-143行目
我々は嚥下筋として二腹筋を分離して検査した[48,49]。また、呼吸筋として横隔膜を分離して検査した[9]。我々は、検査する炎症性サイトカインとしてIL-1b、IL-6、MCP-1を選択し、mRNAを分離して、7、14、21、28日目にその発現レベルを測定した[47]。
6 ページ、図 3A: コントロール/28 日目二腹筋サンプルの p62 ウェスタン ブロット画像のコントラストを下げてください。読者は画像が過度に操作されていると感じる可能性があります。
レビュー担当者のコメントに対する回答:
確かにその通りです。画像のコントラストを下げました。この非常に重要な問題を指摘していただき、ありがとうございます。
7 ページ、214 行目: 「MuRF-1 および Atrogin-1 の mRNA 発現度」を「 MuRF-1 および Atrogin-1発現の経時的プロファイル 」に変更してください。
レビュー担当者のコメントに対する回答:
この非常に重要な問題を指摘していただきありがとうございます。原稿全体で遺伝子名を修正しました。遺伝子の最初の文字を大文字にし、原稿全体で遺伝子をイタリック体にしました。この修正により、原稿は大幅に改善されました。改めて感謝申し上げます。
7 ページ、225 ~ 226 行目: 遺伝子記号「LC3B-II」はやや紛らわしいです。「LC3B」は一般的に使用される遺伝子記号ですが、「LC3-II」は LC3 タンパク質の脂質化形態を指します。
レビュー担当者のコメントに対する回答:
素晴らしいご指導をありがとうございます。原稿全体にわたってご指示どおりに修正させていただきました。
8 ページ、253 行目: 「上記のデータ」を「上記の分子変化」に変更してください。
レビュー担当者のコメントに対する回答:
詳しいコメントをありがとうございます。指示通り変更しました。
8ページ、275行目
上記の分子変化が対照群と 28 日間のチャレンジ群の筋萎縮に及ぼす影響を評価しました。

Reviewer 3 Report
Comments and Suggestions for Authors
The manuscript by Dr. Hashimoto et al is an extensive investigation of the molecular mechanisms linking NRF2 signaling deficiency to muscle atrophy, including the expression levels of proinflammatory cytokines, the activation of calpain/caspase-3 mediated cleavage of myofibrillar proteins, the expression levels of pivotal genes in the ubiquitin-proteosome and autophagy pathways. It is worth praising that the authors also performed in vivo swallow function assessment in mice in addition to muscle histology analysis. The manuscript can be accepted for publication with some minor changes suggested below:
Page 4, line 130-132: Please add “to examine” after “swallowing muscles”, “diaphragms” and “proinflammatory cytokines”.
Page 6, Figure 3A: Please decrease the contrast of the p62 Western blot image for Control/28 day digastric muscle samples. Readers may consider the image to be overmanipulated.
Page 7, line 214: Please change “The mRNA expression degrees of MuRF-1 and atrogin-1” to “The temporal profiles of MuRF-1 and Atrogin-1 expression”.
Page 7, line 225-226: The gene symbol “LC3B-II” is somewhat confusing. “LC3B” is the commonly used gene symbol, while “LC3-II” refers to the lipidated form of LC3 protein.
Page 8, line 253: Please change “above data” to “above molecular changes”.
Author Response
Thank you for your informative and creative comments. Following your comments, the improvements have strengthened and brushed up the manuscript, which the readers will be intensely interested in.
Point-to-point reply to the reviewers
Response to Reviewer 1
Lines 57-64: Since NRF2/KEAP1 signaling plays a key role in this manuscript, the multifaceted role of this pathway deserves to be highlighted. In fact, this signaling plays key functions in the onset and progression of several diseases including cancer (see PMID: 37296999 ).
Response to the reviewer’s comments:
Thank you for your valuable comment. To emphasize the multifaceted role of NRF2, we have added the following sentences.
Page 2, lines 61-62
Nuclear factor erythroid 2-related factor 2 (Nrf2) controls various gene expressions and regulates response to oxidative stress and protection [14-17].
Page 2, lines 67-72
Subsequently, Nrf2 undergoes stabilization and migrates to the nucleus, initiating the activation of numerous genes associated with cellular differentiation, protection, and proliferation [21-24]. Accordingly, Nrf2 plays important roles in cancers [25-27], especially as a target of susceptibility to chemotherapy and radiotherapy [28-30]. In addition, Nrf2 plays potent anti-inflammatory roles, including in the lungs [31-33].
Page 2, lines 76-77
In addition, Nrf2 contributes to the anti-aging process [42].
Figure 1: Higher magnifications are needed to appreciate tissue morphology and staining.
Response to the reviewer’s comments:
You are right. We have added higher magnification images of the lungs to Supplemental Figure 1.
Figure 3: Molecular weights must be added. Correct "a-tubrin" with a-tubulin
Response to the reviewer’s comments:
Thank you for your careful advice. We added the molecular weights as shown previously (ref #53). We have corrected the spelling, as you pointed out.
Figure 7A-C are too small
Response to the reviewer’s comments:
Yes, your comment is totally true. To make it easier for readers to understand, we have retaken the photos and changed them to larger images.
4.4. Western blotting process: Authors must add the product code of all primary antibodies used
I suggest to move the primary antibodies used for WB, IF and IHC in a dedicate table
Response to the reviewer’s comments:
Thank you very much for your suggestion. We followed your instructions and compiled the primary antibodies we used in Supplemental Table 2.
An accurate revision of syntax and terminology is necessary.
Thank you for your in-depth comment. We have corrected the gene names. We capitalized the first letter of the genes and italicized the genes throughout the manuscript. We changed the microtubule-associated protein 1 light chain 3B-II (LC3B-II) abbreviation to Lc3b (Page 7, line 2443). These fixations intensely improved the manuscript; thank you again.
Response to Reviewer 2
- Abstract might be beneficial to include a sentence that briefly summarizes the key findings of the study. This can provide readers with a quick overview of the research.
Response to the reviewer’s comments:
Thank you very much for your helpful advice. We have added it to the abstract as shown below.
Page 1, lines 33-34;
Thus, activation of Nrf2 may alleviate inflammation, muscle atrophy, and function in aspiration pneumonia, an aging population’s major health problem, and may become a therapeutic target.
- I suggest to improve the introduction.
Response to the reviewer’s comments:
Thank you for your valuable comments. We have made substantial revisions to the introduction based on your advice and other reviewers' comments.
- Please, improve the description of study aim and underline the novelty of the study.
Response to the reviewer’s comments:
Thank you for your great suggestion to improve our manuscript. We have added it to the introduction as below.
Page 2, lines 87-91
This study aimed to identify the role of Nrf2 in the exacerbation of pneumonia, muscle atrophy, and swallowing dysfunction in aspiration pneumonia, as well as the role of Nrf2 in its exacerbation mechanism. Thus, we attempted to identify the unknown role of Nrf2 in aspiration pneumonia and the swallowing and respiratory muscles and explored the possibility of Nrf2 as a therapeutic target for the disease.
- Please, underline the most important results to clarify the conclusions.
Response to the reviewer’s comments:
Thank you for the advice. We wrote the most important results in the first paragraph of the discussion section as shown below.
Page 10, lines 343-345
This study demonstrated more exacerbated aspiration pneumonia, swallowing and respiratory muscle atrophy, and swallowing dysfunction in Nrf2-knockouts than in the wild-types, along with their underlying mechanisms.
- The discussion section needs to be improved. It is necessary to clarify the results obtained and compare them with previous or similar studies.
Response to the reviewer’s comments:
Thank you for the critical advice. We added the following sentences.
Page 11, lines 387-394
Previous studies observing lungs showed essential roles of Nrf2 in lung protection from various inflammatory stimulations, such as butylated hydroxytoluene, carrageenin, LPS, and TNF-α [32,41,61]. The Nrf2-knockout mice showed greater lung inflammation and lower survival rate than wild-types [41,61]. These studies showed the effects of Nrf2 in acute lung inflammation. The current study employed chronic lung inflammation and showed greater inflammatory response and muscle atrophy with a lower survival rate in a preliminary experiment in Nrf2-knockout mice than in wild-types. Thus, Nrf2 is essential in lung protection in acute and chronic inflammation.
- I suggest to underline the limitations of the study.
Response to the reviewer’s comments:
Thank you for pointing out this issue. We added the following descriptions.
Page 12, lines 420-427
The limitation of this study. In quantitative RT-PCR, the number of samples that could be detected simultaneously was limited to 96 wells per plate, and we measured each sample in duplicate, which posed a technical limitation. As a result, we could not directly compare the mRNA levels of Nrf2-knockouts and wild-types. We showed mRNA levels of challenged mice in comparison to the controls of wild-types and Nrf2-knockouts, respectively. The current study suggests the protective role of Nrf-2 in lung inflammation. However, we could not overexpress the Nrf2 and confirm its anti-inflammatory effects in this study.
- Please, underline the novelty of the study.
Response to the reviewer’s comments:
Thank you for your great suggestion. The novelty was highlighted as follows.
Page 14, lines 525-526
The new findings in this study suggest that Nrf2 is important for alleviating severe pneumonia, protecting muscles from atrophy, and maintaining muscle function.
- Improve this paragraph and ameliorate the description of statistical tests used to evaluate the data.
Response to the reviewer’s comments:
We are very sorry for our insufficient information in the methods. We added descriptions and improved the methods as shown below.
Page 13, lines 519-525
We showed values as mean values ± standard errors. Body weights were compared at each time point, and fiber sizes were compared at the same fiber size between wild-types and Nrf2-knockouts using a two-tailed paired Student’s t-test. For other values, we statistically analyzed the data using a one-way analysis of variance with Tukey’s multiple comparison test as a post hoc. We employed JMP Pro v. 16.0 software (SAS Institute Inc., Cary, NC) for the data analysis.
Response to Reviewer 3
Page 4, line 130-132: Please add “to examine” after “swallowing muscles”, “diaphragms” and “proinflammatory cytokines”.
Response to the reviewer’s comments:
Thank you for your in-depth comments. We corrected them as shown below.
Page 4, lines 141-143
We isolated the digastric muscles as the swallowing muscles to examine [48,49] and the diaphragms as the respiratory muscles to examine [9]. We chose IL-1b, IL-6, and MCP-1 as proinflammatory cytokines to examine, isolated mRNAs, and measured their expression levels on days 7, 14, 21, and 28 [47].
Page 6, Figure 3A: Please decrease the contrast of the p62 Western blot image for Control/28 day digastric muscle samples. Readers may consider the image to be overmanipulated.
Response to the reviewer’s comments:
You are certainly right. We decreased the contrast of the image. Thank you very much for pointing out this very important issue.
Page 7, line 214: Please change “The mRNA expression degrees of MuRF-1 and atrogin-1” to “The temporal profiles of MuRF-1 and Atrogin-1 expression”.
Response to the reviewer’s comments:
Thank you for pointing out this very important issue. We have corrected the gene names all through the manuscript. We made the first letter of the genes a capital and italicized the genes all through the manuscript. This fixation intensely improved the manuscript; thank you again.
Page 7, line 225-226: The gene symbol “LC3B-II” is somewhat confusing. “LC3B” is the commonly used gene symbol, while “LC3-II” refers to the lipidated form of LC3 protein.
Response to the reviewer’s comments:
Thank you for your great instruction. We have corrected it as instructed all through the manuscript.
Page 8, line 253: Please change “above data” to “above molecular changes”.
Response to the reviewer’s comments:
Thank you for your in-depth comment. We made the changes as instructed.
Page 8, line 275
We assessed the effects of the above molecular changes on muscle atrophy in the control group and the 28-day challenge group.

Round 2
Reviewer 1 Report
Comments and Suggestions for Authors
the manuscript has been significantly improved and can be accepted in the present form
Author Response
Your comments greatly improved our manuscript; thank you.

Reviewer 2 Report
Comments and Suggestions for Authors
The manuscript has been improved. I suggest authors to ameliorate the conclusions:
1) 5. Conclusions 511 This research assessed the involvement of Nrf2 in aspiration pneumonia and the 512 muscles for swallowing and respiration in a preclinical animal model. The new findings 513 in this study suggest that Nrf2 is important for alleviating severe pneumonia, protecting 514 muscles from atrophy, and maintaining muscle function. Since aspiration pneumonia is 515 widespread and fatal in an aging society, Nrf2 may have the potential to become a novel 516 target for protecting patients from seriously ill conditions. Please, improve this part and underline the Future prospects
Author Response
The manuscript has been improved. I suggest authors to ameliorate the conclusions:
1) 5. Conclusions 511 This research assessed the involvement of Nrf2 in aspiration pneumonia and the 512 muscles for swallowing and respiration in a preclinical animal model. The new findings 513 in this study suggest that Nrf2 is important for alleviating severe pneumonia, protecting 514 muscles from atrophy, and maintaining muscle function. Since aspiration pneumonia is 515 widespread and fatal in an aging society, Nrf2 may have the potential to become a novel 516 target for protecting patients from seriously ill conditions. Please, improve this part and underline the Future prospects
Response to the reviewer’s comment:
Thank you for your creative suggestion. Following your suggestion, we ameliorated the conclusion below, written in blue characters in the re-revised version.
Page 14, lines 512-520
This research assessed the involvement of Nrf2 in aspiration pneumonia, followed by atrophy and dysfunction of aspiration-related muscles in a preclinical animal model. The new findings in this study suggest that Nrf2 is important for alleviating severe pneumonia, protecting muscles from atrophy, and maintaining muscle function. Since aspiration pneumonia is widespread and fatal in an aging society, Nrf2 may have the potential to become a novel target for protecting patients from seriously ill conditions. Indeed, there is a worldwide competition to develop Nrf2 activators, and some are currently approved for Friedreich ataxia and multiple sclerosis [71,72]. A future perspective may involve the clinical trials of Nrf2 activators in aspiration pneumonia.
